# FuseUNet: A Multi-Scale Feature Fusion Method for U-like Networks

**Quansong He** [* 1]   **Xiangde Min** [* 2]   **Kaishen Wang** [3]   **Tao He** [1]

## Abstract

Medical image segmentation is a critical task in computer vision, with UNet serving as a milestone architecture. The typical component of UNet family is the skip connection, however, their skip connections face two significant limitations: (1) they lack effective interaction between features at different scales, and (2) they rely on simple concatenation or addition operations, which constrain efficient information integration. While recent improvements to UNet have focused on enhancing encoder and decoder capabilities, these limitations remain overlooked. To overcome these challenges, we propose a novel multi-scale feature fusion method that reimagines the UNet decoding process as solving an initial value problem (IVP), treating skip connections as discrete nodes. By leveraging principles from the linear multistep method, we propose an adaptive ordinary differential equation method to enable effective multi-scale feature fusion. Our approach is independent of the encoder and decoder architectures, making it adaptable to various U-Net-like networks. Experiments on ACDC, KiTS2023, MSD brain tumor, and ISIC2017/2018 skin lesion segmentation datasets demonstrate improved feature utilization, reduced network parameters, and maintained high performance. The code is available at `https://github.com/nayutayuki/FuseUNet`.

## 1. Introduction

Medical image segmentation is a crucial branch of computer vision, and with the development of deep learning, many excellent segmentation methods have emerged. UNet (Ronneberger et al., 2015) is a milestone network architecture in this field, characterized by its symmetric encoder-decoder convolutional network structure and the skip connections for integrating features from different scales.

The emergence of UNet laid the foundation for segmentation networks, influencing the design of many existing architectures. For simplicity, we refer to these networks as "U-Nets". Recently popular U-Nets improvement strategies have primarily focused on enhancing the information processing capabilities of the encoder and decoder, such as UNETR (Hatamizadeh et al., 2021), Swin-UNet (Cao et al., 2022), and their variants like Swin-UNETR (Tang et al., 2022), MetaUNETR (Lyu et al., 2024) which incorporate Transformer (Vaswani, 2017); VM-UNet (Ruan & Xiang, 2024), UltraLight VM-Unet (Wu et al., 2024), and LKM-UNet (Wang et al., 2024a) which incorporate Mamba (Gu & Dao, 2023); and Rolling-UNet (Liu et al., 2024), DHMF-MLP (Cheng & Wang, 2023), which improve multi-layer perceptrons (MLP). However, the skip connections in U-Nets are limited to feature fusion within the same scale, lacking interaction across different scales, which is a clear limitation. Additionally, the feature fusion strategy in U-Nets relies solely on simple addition or concatenation, which makes it difficult to effectively integrate feature information.

Unfortunately, there has been little research on skip connections in recent years. The most recent notable studies addressing this issue are UNet++ (Zhou et al., 2020) and UNet3+ (Huang et al., 2020). UNet++ introduced many intermediate nodes and dense skip connections, using feature summation to integrate features from different scales. UNet3+ built on UNet++ by proposing a full-scale skip connection, where each decoder processes all scales of skip connections. Their experimental results demonstrate the benefits of adding multi-scale information interaction in the UNet structure. In fact, the idea behind U-Nets is quite analogous to the numerical solution of ordinary differential equation (ODE). In U-Nets, the decoder simulates functional relationships based on multiple known encoder output feature nodes to obtain the corresponding results. This process is similar to computing approximate values of a function at discrete points in ODE. However, the aforementioned networks still follow UNet's approach for feature fusion, which relies on simple concatenation or addition. This straightforward method is analogous to the explicit

---

*Equal contribution  [1]College of Computer Science, Sichuan University, Chengdu, China  [2]Tongji Hospital Tongji Medical College, Huazhong University of Science and Technology, Wuhan, China  [3]Department of Computer Science, University of Maryland, College Park, America.  Correspondence to: Tao He <tao_he@scu.edu.cn>.

*Proceedings of the 42nd International Conference on Machine Learning*, Vancouver, Canada. PMLR 267, 2025. Copyright 2025 by the author(s).

Euler method (Euler, 1845) in mathematics, which has only first-order accuracy and fails to fully utilize the available feature information. To better leverage known information for feature fusion, we propose using the linear multistep method (Bashforth & Adams, 1883; Moulton, 1928), a widely used mathematical tool, in this paper.

In recent years, various discretization methods (He et al., 2024; Wang et al., 2024c; Niu et al., 2024; He et al., 2023) involving neural memory ordinary differential equations (nmODEs) (Yi, 2023) has been developed, showing some potential of the linear multistep method in image segmentation tasks. These methods started by designing the decoder, enabling information interaction between adjacent stages, which helps in feature fusion. However, the discrete methods they use have at most second-order accuracy, rely only on information between adjacent stages, and lack the capability for multi-scale information interaction.

According to the mathematical principles of the linear multistep method, we propose a novel multi-scale feature fusion method for skip connections: treating the feature information of skip connections at various scales as a sequence and viewing the decoding process of U-Nets as an initial value problem (IVP). We introduce the nmODEs and adapt them to this problem by designing an adaptive high-order discretization method that discretizes the process. This method processes the sequence of skip connections using discrete nmODEs and ultimately generates the segmentation map. This method is relied on skip connections and is not limited to the types of encoders and decoders, making it theoretically applicable to a wide range of U-Net variants.

To demonstrate the effectiveness and generalizability of the proposed method, we applied it to three mainstream U-Nets based on convolution, Transformer, and Mamba architectures. Experiments on 3D tasks (ACDC, KiTS2023, MSD brain tumor) as well as 2D tasks (ISIC2017 and ISIC2018 skin lesion segmentation tasks), show that the proposed method significantly improves the utilization efficiency of the information extracted by encoders, while drastically reducing network parameters and maintaining network performance.

## 2. Related Work

### 2.1. UNet and U-Nets

UNet (Ronneberger et al., 2015), as the foundation of all U-Nets, features a symmetric encoder-decoder structure with skip connections that facilitate information flow between them. Based on its structural characteristics, improvements to UNet generally follow two main directions: optimizing the connection strategy within the network architecture or introducing more efficient modules in the encoders and decoders.

In the first direction, previous studies primarily focused on increasing connection density. ResUNet (Xiao et al., 2018) and DenseUNet (Cai et al., 2020) achieved this by incorporating residual and dense connections within the encoder and decoder. UNet++ (Zhou et al., 2020) introduced numerous intermediate nodes and adopted dense skip connections, while UNet3+ (Huang et al., 2020) further enhanced UNet++ by designing more comprehensive full-scale skip connections. In recent years, research in this area has received less attention due to the rise of advanced modules such as Transformer (Vaswani, 2017), shifting the focus towards the second direction.

A representative approach in the second direction is the introduction of attention mechanisms. For instance, TransUNet (Chen et al., 2024) and UNETR (Hatamizadeh et al., 2021) designed encoder architectures based on Transformer, while retaining convolutional decoders. Swin-UNet (Cao et al., 2022), on the other hand, used a pure Transformer approach, and Swin-UNETR (Tang et al., 2022) built on UNETR by introducing a sliding attention mechanism. MetaUNETR (Lyu et al., 2024) proposed the TriCruci module in an attempt to create a unified segmentation framework. Additionally, the Mamba (Gu & Dao, 2023) architecture gained attention due to its lower complexity compared to Transformer. LKM-UNet (Wang et al., 2024a) demonstrated the feasibility and effectiveness of using large Mamba kernels to achieve a large receptive field, and UltraLight VM-Unet (Wu et al., 2024) introduced a new Parallel Vision Mamba layer on top of VM-UNet (Ruan & Xiang, 2024), significantly reducing the number of parameters and computational load in skin lesion segmentation tasks. Furthermore, research on MLP (Liu et al., 2024; Cheng & Wang, 2023) and other mechanisms in segmentation models has been observed, although these are not the mainstream methods. However, all these networks share a common limitation: the lack of information interaction across different scales in their skip connections.

### 2.2. Neural Ordinary Differential Equations

Ordinary differential equation (ODE) systems, a key class of dynamical systems, have been widely studied in mathematics and physics. Neural ODEs (NODEs) (Chen et al., 2018) offered a mathematical framework to interpret ResNet, transforming it from a black box into a comprehensible model by viewing neural networks as ODE representations.

However, NODEs face limitations, as models using data as initial values can only learn features within the same topological space (Dupont et al., 2019). Furthermore, attractors in dynamical systems are linked to memory capacity (Poucet & Save, 2005; Wills et al., 2005), but conventional NODEs cannot fully exploit this capacity. To address these limitations, nmODEs (Yi, 2023) extended NODEs by en-

hancing nonlinear expression through implicit mapping and nonlinear activation functions. Unlike traditional NODEs, nmODEs treat inputs as external parameters, not initial values. They separate the neuron's function into learning and memory components: learning happens in the learning part, while the memory part maps inputs to global attractors, linking input space to memory space.

nmODEs have been successfully applied to various segmentation tasks. For example, BiFNN (Niu et al., 2024) employed bidirectional skip connections and a nonparametric backward path based on nmODEs, which improved image recognition performance over models such as ResNet and Vision Transformer. nmPLS-Net (Dong et al., 2023) utilized the nonlinear representation and memory capabilities of nmODEs for edge-based decoding, achieving precise lung lobe segmentation. Incorporating nmODEs into UNet through simple discretization has also shown promising results in diabetic kidney (Wang et al., 2024b) and skin cancer lesion segmentation (He et al., 2024; 2023; Wang et al., 2024c). Furthermore, nmODEs have improved the robustness of medical image segmentation (Hu et al., 2023), demonstrating their versatility and effectiveness across diverse medical segmentation challenges.

## 3. Methods

### 3.1. High-order Discrete Methods

The traditional structure of U-Nets, as shown in Fig. 1, clearly indicates that the skip connections only communicate information within the same stage. Consider a U-Net with $L$ stages, for example, in the classic UNet, $L = 5$. The skip connection at the same stage of the encoder is denoted as $X_i$, and the output of the decoder is denoted as $Y_i$. The abstracted mathematical model can be expressed as: $Y_i = f(X_i, Y_{i-1})$. In this formulation, only the result from the previous step $Y_{i-1}$ and the features from the current step $X_i$ are used. This single-step computation method is simple and intuitive, but it fails to leverage information from previous stages of $X_i$ and the results obtained before $Y_{i-1}$, and the accuracy of this method is limited. To fully utilize the information from the skip connections and improve accuracy, a natural next step is to adopt a multistep approach. This would allow the network to incorporate information from both the past and current steps, providing more comprehensive feature integration and ultimately enhancing performance.

The linear multistep method is a classical numerical method that uses information from the current time step and several previous time steps to predict the solution at the next time step. This method aligns perfectly with the idea of fully utilizing the skip connection information $X_i$ and the computed values $Y_i$ from U-Nets. The theorem of linear multistep method is given as follow:

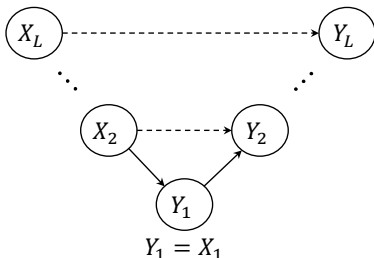

*Figure 1.* The traditional architecture of U-Nets, the skip connections only communicate information at the same scale.

method is given as follow:

**Theorem 3.1.** *Linear Multistep Method (Bashforth & Adams, 1883; Moulton, 1928). Given the derivative $\dot{y}(t) = F(t, y(t)), y(t_0) = y_0$, choose a value $\delta$ for the size of every step along t-axis and set $t_{n+i} = t_n + i \cdot \delta$, the result is approximations for the value of $y(t_i) \approx y_i$, multistep methods use information from the previous $s$ steps to calculate the next value:*

$$\sum_{j=0}^{s} a_j y_{n+j} = \delta \sum_{j=0}^{s} b_j F(t_{n+j}, y_{n+j}), \quad (1)$$

*with $a_s = 1$. The coefficients $a_0, \ldots, a_{s-1}$ and $b_0, \ldots, b_s$ determine the method.*

In **Theorem** 3.1, if $b_s = 0$, then the method is called "explicit", since the formula can directly compute $y_{n+s}$. If $b_s \neq 0$ then the method is called "implicit", since the value of $y_{n+s}$ depends on the value of $F(t_{n+s}, y_{n+s})$, and the equation must be solved for $y_{n+s}$.

*Table 1.* Linear multistep method.

| step | order | equation |
|------|-------|----------|
| Explicit: Adams-Bashforth (AB) Method | | |
| 1 | 1 | $y_{n+1} = y_n + \delta \cdot F_n$ |
| 2 | 2 | $y_{n+2} = y_{n+1} + \frac{\delta}{2} \cdot (3F_{n+1} - F_n)$ |
| 3 | 3 | $y_{n+3} = y_{n+2} + \frac{\delta}{12} \cdot (23F_{n+2} - 16F_{n+1} + 5F_n)$ |
| 4 | 4 | $y_{n+4} = y_{n+3} + \frac{\delta}{24} \cdot (55F_{n+3} - 59F_{n+2} + 37F_{n+1} - 9F_n)$ |
| Implicit: Adams-Moulton (AM) Method | | |
| step | order | equation |
| 1 | 2 | $y_{n+1} = y_n + \frac{\delta}{2} \cdot (F_n + F_{n+1})$ |
| 2 | 3 | $y_{n+2} = y_{n+1} + \frac{\delta}{12} \cdot (5F_{n+2} + 8F_{n+1} - F_n)$ |
| 3 | 4 | $y_{n+3} = y_{n+2} + \frac{\delta}{24} \cdot (9F_{n+3} + 19F_{n+2} - 5F_{n+1} + F_n)$ |

Increasing the number of steps (i.e., incorporating more previous time points) raises the order of the linear multistep method, thereby improving its theoretical accuracy. However, it may also reduce stability, cause error accumulation, and increase dependence on initial values. Therefore, the highest order typically used for the linear multistep method is 4. For simplicity, we write $F(t_n, y_n)$ as $F_n$, the specific coefficients for the linear multistep method are shown in

Table 1. Besides, the explicit method is denoted as $AB_i$ and the implicit method as $AM_i$ in section 3.2, where $i$ corresponds to the step number in Table 1.

When the number of steps is the same, implicit methods consistently achieve higher accuracy than explicit methods. Consequently, we strive to use implicit methods whenever possible. However, implicit methods require the derivative values at the current node, which are unknown at the current step. The predictor-corrector method provides a mathematical solution to this problem. The corresponding theorem is presented as follows:

**Theorem 3.2.** *Predictor-Corrector Method (Heun, 1900; Gragg & Stetter, 1964). Consider the differential equation $\dot{y}(t) = F(t, y(t)), y(t_0) = y_0$, and denote the step size by $\delta$. First, starting from the current value $y_i$, calculate an initial guess value $\overline{y_{i+1}}$ via an explicit method. Next, improve the initial guess using corresponding implicit method. For example use 1 step methods:*

$$\overline{y_{n+1}} = y_n + \delta \cdot F(t_n, y_n)$$
$$y_{n+1} = y_n + \frac{\delta}{2} \cdot (F(t_n, y_n) + F(t_{n+1}, \overline{y_{n+1}})), \tag{2}$$

*where $y_n \approx y(t_n)$. $\overline{y_{n+1}}$ is the predicted median, $y_{n+1}$ is the final result corrected for $\overline{y_{n+1}}$.*

Thus, more accurate solutions can be obtained using implicit methods.

### 3.2. Adaptive Discrete Method for U-Nets

The linear multistep method is a mathematical tool used to solve initial value problem (IVP) in ordinary differential equation (ODE). To relate the architecture of U-Nets with ODE, existing methods typically introduce differential equations starting from the decoder. Following this idea, we define the differential relationship between the skip connections $X$ and the corresponding stages $Y$ as $F$, but we focus on fusing multi-scale information starting from the skip connections. The function $F$ will be explained in detail in section 3.3. In this section, we first describe the proposed discrete method and the overall framework applied in an L-stage U-Net, as shown in Fig. 2 (a).

First, we map the data from each stage of the U-Net to the elements of the linear multistep method: we treat the process where the decoder reconstructs the features extracted by the encoder into predicted labels as solving an IVP. $X_i$ represents a series of discrete nodes, and $Y_i$ represents the corresponding solution at each node, $1 \leq i \leq L$. $Y_{final}$ is the solution to the IVP. The bottom stage represents the start of the IVP, where we initialize $Y_1 = 0$, representing an empty memory stream. As the IVP progresses, we gradually fill in information on the empty memory stream, first mapping the coarse high-level features to the feature map, and then filling in the finer low-level details.

---

**Algorithm 1** Adaptive discrete method for U-Nets

**NOTE:** $F_{1:i}$ represents the sequence $\{F_1, F_2, \ldots, F_i\}$, indicating all values from $F_1$ to $F_i$.
**Input:** feature map $X$, memory statement $Y$ and total stage $L$
**for** $i = 1$ **to** $L - 1$ **do**
  **if** $i < 4$ **then**
    **Predictor:** $Y_{i+1} = AB_i(Y_i, F_{1:i})$
    **Corrector:** $Y_{i+1} = AM_i(Y_{i+1}, F_{1:i+1})$
  **else**
    **Predictor:** $Y_{i+1} = AB_4(Y_i, F_{i-3:i})$
    **Corrector:** $Y_{i+1} = AM_3(Y_{i+1}, F_{i-2:i+1})$
  **end if**
**end for**
**if** $L < 5$ **then**
  $Y_{final} = AB_L(Y_L, F_{1:L})$
**else**
  $Y_{final} = AB_4(Y_L, F_{L-3:L})$
**end if**
**Return** $Y_{final}$

---

Then, we can apply the mathematical ideas of the linear multistep method to process features at multiple scales. To ensure effective interaction between features at different scales, it is crucial to incorporate as much information from previous stages as possible and employ high-accuracy implicit methods when deriving the output of a given decoder stage. When the current stage number is greater than 4 and not the last stage, a 4-step implicit method can be straightforwardly chosen for prediction-correction. However, the linear multistep method need to calculates the values of $Y_l, l < i$ for the previous steps using a single-step method first, and then substitutes the calculated derivatives into the formula. The single-step method at the initialization step is still limited to the same scale, which does not align with our goal of multi-scale feature interaction. Therefore, we need to modify the initialization step of the linear multistep method. Our approach is to repeatedly apply the lower-step linear multistep method during the startup phase of the high-step method. Specifically, when calculating $y_{i+1}$, we use the $i$-step implicit method. For $i > 4$, the 4-step implicit method is applied until the $L$-th stage, where the final computation is performed using an explicit method. This algorithm is outlined in **Algorithm 1**, and the detailed computational process is provided in Appendix A.

Finally, we apply a convolution layer to map $Y_{final}$ to the segmentation map.

### 3.3. Differential Equations Adapted for U-Nets

nmODEs divided neuron into two parts: learning neuron and memory neuron. The input data is passed to the learning

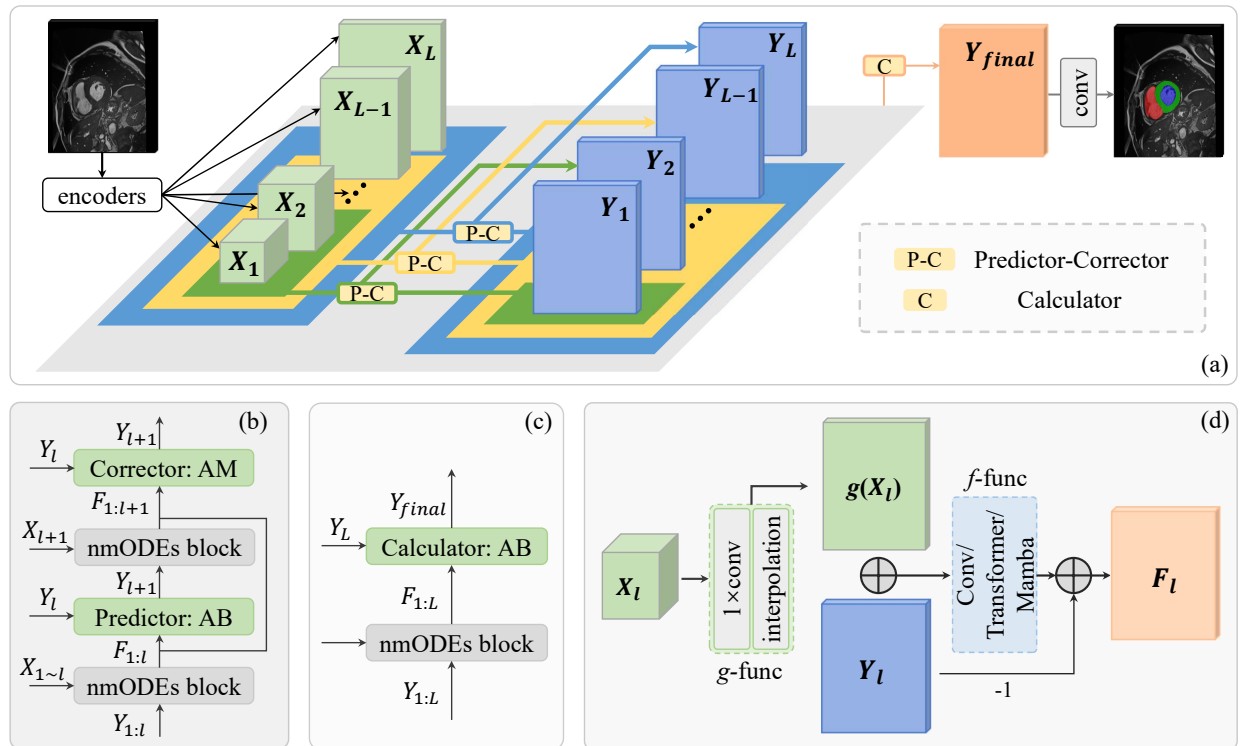

*Figure 2.* (a) The architecture of an $L$-stage U-Net incorporating discrete nmODEs. Here, $P$-$C$ represents the Predictor-Corrector module, with its internal structure detailed in (b). $C$ denotes the calculator used in the final step, which exclusively employs explicit methods, and its internal structure is illustrated in (c). The number of channels in $Y$ is set to twice the number of target classes, while all other dimensions remain consistent with the original input. The internal structure of the nmODEs block is shown in (d), where the function $f$ executes the corresponding operations based on the specific network architecture, such as convolution, Transformer, or Mamba.

neuron to extract features, while the memory carrier records the extracted information in the memory neuron. This design has three advantages. nmODEs inherently avoids the features learned from being isomorphic to the input data space. Additionally, we have identified two other advantages that are particularly relevant to this work:

First, the separation of learning and memory neurons in nmODEs allows it to retain a continuous memory flow while still maintaining the ability to process external inputs. Therefore, features extracted by the encoders from different scales can be fed into the learning neuron of nmODEs as a sequence of nodes, and the memory carrier is updated with the numerical solutions corresponding to each node.

Second, the memory carrier ultimately outputs the target data but does not participate in the feature extraction process. As a result, it does not need to retain much information other than the final output. This reduces the required number of channels significantly compared to the decoder of traditional U-Nets, thus greatly reducing computational costs.

The equation for nmODEs is given as follows:

$$\dot{Y_t} = -Y_t + f\left(Y_t + g(X_t)\right), \tag{3}$$

where $X_t$ represents the external input at time $t$, $Y_t$ represents the memory flow carried by the memory neuron at time $t$, with the initial condition $Y_0 = 0$ representing an empty memory flow, $f(\cdot)$ is a non-linear mapping that combines the existing memory and the new external input to update the memory flow using a differential equation numerical solver, $g(\cdot)$ is the function that processes the external input $X_t$ at each time step. When applying nmODEs to the discrete U-Nets, the following equation holds:

$$\dot{Y_i} = -Y_i + f\left(Y_i + g(X_i)\right). \tag{4}$$

In U-Nets, the $X_i$ and $Y_i$ in Eq. (4) correspond to those in Section 3.2. The function $g(\cdot)$ uses convolution and interpolation to align the channels and shape of $X_i$ and $Y_i$, while the choice of $f(\cdot)$ depends on the network to which this equation is applied. By substituting Eq. (3) into the linear multistep method proposed in **Algorithm 1**, we can obtain the approximate solution for each stage. Following this process, we designed three modules: Predictor-Corrector, Calculator, and the nmODEs block, as shown in Fig. 2 (b), (c), and (d), respectively.

*Table 2.* Overview of Used Datasets.

| Dataset | Region | Data Type | Number of Samples | Target Classes |
|---|---|---|---|---|
| ACDC (Bernard et al., 2018) | Heart | 3D MRI | 100 | 3 (1-MYO, 2-RV,3-LV) |
| KiTS23 (Heller et al., 2023) | Kidney | 3D CT | 559 | 3 (1-Kidney, 2-Cyst, 3-Tumor) |
| MSD (Antonelli et al., 2022) | Brain | 3D MRI | 484 | 3 (1-WT, 2-ET, 3-TC) |
| ISIC2017 (Codella et al., 2018a) | Skin | 2D Dermoscopy | 2000 | 1 (Lesion skin area) |
| ISIC2018 (Codella et al., 2018b) | Skin | 2D Dermoscopy | 2594 | 1 (Lesion skin area) |

## 4. Experiments

The currently popular U-Nets are primarily based on three architectures: convolution, Transformer, and Mamba. To evaluate the effectiveness and generalizability of the proposed method, we select a representative backbone network from each of these architectures and perform experiments on the datasets used in their respective original studies. All experiments were conducted on a single RTX 4090. Except for the learning rate, all experimental settings were consistent with those of the backbone networks used for comparison. Due to the significant reduction in the number of parameters, the learning rate was set to 2 or 3 times that of the backbone network's setting. The detailed hyperparameter settings are provided in Appendix B.

### 4.1. Datasets and Backbone Networks

Table 2 presents the details of the datasets used in this paper. In the table, LV denotes the left ventricle, RV represents the right ventricle, MYO corresponds to the myocardium, EC indicates the enhancing tumor, TC refers to the tumor core, and WT denotes the whole tumor.

**nn-UNet.** For CNN-based U-Nets, nn-UNet (Fabian et al., 2021; Isensee et al., 2024) remains the backbone due to its superior applicability, outperforming nearly all other U-Nets across reliable datasets. **UNETR.** In Transformer-based U-Nets, UNETR (Hatamizadeh et al., 2021), from the MONAI framework, is a key foundation for research and a suitable backbone for our study. **UltraLight VM-UNet.** Mamba's efficiency and linear complexity reduce costs over Transformers. UltraLight VM-UNet (Wu et al., 2024) leverages this, achieving strong performance with a lightweight design. We use it to assess our method's potential for further downsizing.

### 4.2. Main Results

**3D Tasks.** All 3D tasks report Dice scores (%) using five-fold cross-validation, following the protocol of the backbone network. The performance data for the compared networks are sourced from (Hatamizadeh et al., 2021; Isensee et al., 2024; Perera et al., 2024). Table 3 presents the performance of the proposed method on 3D tasks. The gray-shaded rows in the table indicate FuseUNet and its corresponding backbone network. When using the convolutional nn-UNet as the backbone, FuseUNet achieves a 54.9% reduction in the number of parameters and a 34.3% reduction in GFLOPs, while maintaining the performance of nn-UNet. Although its performance on the ACDC dataset is slightly lower, it surpasses nn-UNet on the larger KiTS dataset. Detailed performance data on each fold is provided in Appendix D.

When using the Transformer-based UNETR as the backbone network, FuseUNet achieves a 13.6% reduction in the number of parameters and a 50% reduction in GFLOPs, while surpassing UNETR's performance by 1.5%. The smaller reduction in parameters compared to nn-UNet is due to UNETR's parameters being primarily concentrated in the encoder's attention module, with the decoder contributing a smaller proportion. Since FuseUNet's multi-scale feature interaction method is designed around skip connections and does not modify the encoder, the reduction in the number of parameters is relatively modest.

Fig. 3 illustrates the visual segmentation results of Fuse-UNet, where (a), (b), and (c) correspond to the ACDC, KiTS, and MSD brain tumor datasets, respectively. In Figure 3 (a), nn-UNet frequently makes errors in identifying the right ventricle (RV), sometimes misclassifying unrelated tissues as the RV or missing portions of it entirely. FuseUNet significantly mitigates these issues. Fig. 3(b) shows that nn-UNet occasionally fails to fully recognize the kidneys and often confuses kidney tumors with cysts. This is particularly evident in the third row, where nn-UNet incorrectly classifies most cysts as tumors, even in the absence of tumor tissue. FuseUNet greatly improves upon this. In Fig. 3(c), UNETR exhibits challenges such as incomplete tumor recognition and missing enhanced tumor regions, both of which are notably addressed by FuseUNet.

**2D Tasks.** For 2D tasks, we report Dice, sensitivity, specificity, and accuracy as percentages, following the protocol of the backbone network. The performance data for the compared networks are sourced from (Wu et al., 2024). Table 4 shows the performance of the proposed method on 2D tasks, using the lightweight UltraLight VM-UNet with a Mamba architecture as the backbone network. While FuseUNet reduces the number of parameters by 29.8%, its GFLOPs have increased by 0.075, with performance comparable to that of UltraLight VM-UNet. This increase in GFLOPs is

*Table 3.* The performance comparison between FuseUNet, the backbone networks, and SOTA on 3D tasks.

| Dataset | Model | Params(M) | GFLOPs | Dice1 | Dice2 | Dice3 | Dice avg |
|---------|-------|-----------|--------|-------|-------|-------|----------|
| ACDC | CoTr (Xie et al., 2021) | 41.9 | 668.1 | 89.06 | 88.56 | 94.06 | 90.56 |
|  | Swin-UNETR (Tang et al., 2022) | 62.8 | 384.2 | 89.56 | 89.98 | 94.33 | 91.29 |
|  | U-Mamba (Ma et al., 2024) | 173.5 | 1255 | 89.71 | 89.70 | 94.25 | 91.22 |
|  | STU-Net-L (Huang et al., 2023) | 440.3 | 3810 | 90.02 | 89.59 | 94.32 | 91.31 |
|  | nn-UNet (Isensee et al., 2024) | 31.2 | 402.6 | 90.11 | 89.96 | **94.55** | 91.54 |
|  | FuseUNet (Ours) | **14.0** | **264.9** | **90.18** | **90.05** | 94.49 | **91.57** |
| KiTS | nnFormer (Zhou et al., 2022) | 150.1 | 425.8 | 92.27 | 69.78 | 65.53 | 75.86 |
|  | Swin-UNETR (Tang et al., 2022) | 62.8 | 384.2 | 94.48 | 76.80 | 72.53 | 81.27 |
|  | U-Mamba (Ma et al., 2024) | 173.5 | 1255 | 96.08 | 82.84 | **79.77** | **86.23** |
|  | STU-Net-L (Huang et al., 2023) | 440.3 | 3810 | 96.11 | 82.35 | 79.09 | 85.85 |
|  | nn-UNet (Isensee et al., 2024) | 31.2 | 402.6 | 96.03 | 82.65 | 79.44 | 86.04 |
|  | FuseUNet (Ours) | **14.0** | **264.9** | **96.47** | **83.06** | 79.04 | 86.19 |
| MSD | UNet (Ronneberger et al., 2015) | 13.4 | 31.1 | 76.6 | 56.1 | 66.5 | 66.4 |
|  | UNet3+ (Huang et al., 2020) | **12.0** | - | 62.2 | 41.4 | 47.8 | 50.5 |
|  | TransUNet (Chen et al., 2024) | 116.5 | - | 70.6 | 54.2 | 68.4 | 64.4 |
|  | Swin-UNETR (Tang et al., 2022) | 62.8 | 384.2 | 70.0 | 52.6 | 70.6 | 64.4 |
|  | UNETR (Hatamizadeh et al., 2021) | 103.7 | 40.3 | 78.9 | 58.5 | 76.1 | 71.1 |
|  | FuseUNet (Ours) | 89.2 | 20.1 | **79.5** | **60.1** | **78.2** | **72.6** |

(a) Visualization on the ACDC    (b) Visualization on the KiTS    (c) Visualization on the MSD

*Figure 3.* The visual comparison of segmentation results between FuseUNet and the backbone networks on 3D tasks.

attributed to the interpolations used by FuseUNet to align the shapes of skip connection $X_i$ and the corresponding $Y_i$. This issue is less noticeable when the backbone network has a larger GFLOPs, but it becomes more pronounced in lightweight networks. However, overall, it only leads to a minor increase in GFLOPs at the decimal level.

Fig. 4 illustrates the visual segmentation results of Fuse-UNet, where (a) and (b) correspond to the ISIC2017 and ISIC2018 datasets, respectively. FuseUNet addresses the false negative and false positive issues frequently observed in UltraLight VM-UNet. Moreover, it captures lesion boundary details that are significantly closer to the ground truth.

This subsection presents only a few visualization results, with additional results provided in Appendix E.

### 4.3. Ablation Experiments

**The impact of the number of feature fusion steps.** This experiment was conducted on the first fold of all 3D datasets and the full set of 2D datasets. Figure 5 illustrates the performance (Dice) when different maximum orders of the linear multistep method are used. Since the baseline varies across datasets, we have normalized the data. As the order increases, more skip connections participate in feature information interaction. The results indicate a strong correlation between the network's performance and the highest order of the applied feature interaction method.

**The impact of memory capacity.** This experiment was conducted on the first fold of the KiTS dataset. Table 5 shows the impact of channel count in $Y$ on performance, where $N$ is the number of target classes. Setting $Y$ to $2N$

Table 4. The performance comparison between FuseUNet, the backbone networks and SOTA on 2D tasks.

| Model | Params (M) | GFLOPs | ISIC2017 | | | | ISIC2018 | | | |
|---|---|---|---|---|---|---|---|---|---|---|
| | | | Dice | SE | SP | ACC | Dice | SE | SP | ACC |
| UNet (Ronneberger et al., 2015) | 13.4 | 31.12 | 89.89 | 87.93 | 98.12 | 96.13 | 88.51 | 87.35 | **97.44** | 95.47 |
| TransFuse (Zhang et al., 2021) | 26.16 | 11.5 | 79.21 | 87.14 | 97.98 | 96.17 | 89.27 | **91.28** | 95.74 | 94.66 |
| MALUNet (Ruan et al., 2022) | 0.177 | 0.085 | 88.96 | 88.24 | 97.62 | 95.83 | 89.31 | 88.90 | 97.25 | 95.48 |
| EGE-UNet (Ruan et al., 2023) | 0.053 | 0.072 | **90.73** | 89.31 | 98.16 | 96.42 | 88.19 | 90.09 | 96.38 | 95.10 |
| VM-UNet (Ruan & Xiang, 2024) | 27.427 | 4.112 | 90.70 | 88.37 | **98.42** | 96.45 | 88.91 | 88.09 | 97.43 | 95.54 |
| UltraLight VM-UNet (Wu et al., 2024) | 0.049 | **0.060** | 90.64 | 88.85 | 98.38 | **96.63** | 89.40 | 86.80 | 96.38 | 95.10 |
| FuseUNet (Ours) | **0.036** | 0.095 | 90.69 | **89.59** | 98.20 | 96.62 | **89.77** | 89.10 | 97.41 | **95.62** |

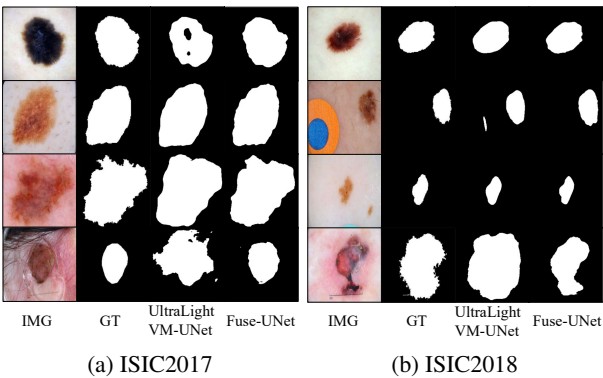

(a) ISIC2017      (b) ISIC2018

Figure 4. The visual comparison of segmentation results between FuseUNet and the backbone networks on 2D tasks.

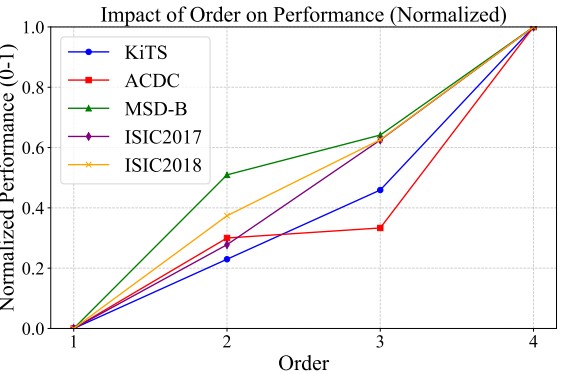

Figure 5. The impact of the number of feature fusion steps on network performance.

significantly outperforms $N$, suggesting that some redundancy in memory flow is beneficial. However, $3N$ reduces performance, and $4N$ offers only a slight gain, indicating excessive redundancy is unnecessary. As the linear multistep method requires storing multiple intermediate states, memory consumption increases with channel count. Using more than $4N$ is inefficient, making $2N$ the optimal balance of performance and cost. Thus, all subsequent experiments adopt $2N$ for $Y$.

Table 5. The impact of memory capacity.

| Channels | GFLOPs | VRAM(G) | Epoch(s) | Dice |
|---|---|---|---|---|
| N | 264 | 5.2 | 125 | 85.86 |
| 2N | 265 | 5.9 | 128 | 86.74 |
| 3N | 266 | 7.1 | 135 | 86.63 |
| 4N | 267 | 8.7 | 147 | 86.88 |

## 5. Conclusion

This paper reinterprets skip connections in U-Nets from a numerical computation perspective for the first time. We propose a multi-scale feature fusion method that integrates the linear multistep method, predictor-corrector method, and nmODEs, dynamically selecting the optimal discretization order for efficient cross-scale fusion. As a skip connection strategy, it is encoder-decoder agnostic and theoretically applicable to any U-Nets. Experiments validate its effectiveness across convolution-, Transformer-, and Mamba-based U-Nets, reducing computational costs while preserving backbone performance. Visualizations further highlight improved edge recognition and feature distinction. Our work moves beyond empirical network design, and reveals the underlying connection between skip connections and numerical integration methods. This demonstrates that classical numerical computation theory can provide an interpretable mathematical foundation for network structures, offering a new perspective on the cross-layer information propagation mechanism in U-Nets. However, the method's reliance on the linear multistep principle introduces a limitation: the need to store multiple historical solutions, leading to high memory consumption, particularly in large-scale problems with numerous target categories. Addressing this challenge remains a key area for future research.

## Acknowledgements

This work was supported by the National Natural Science Foundation of China under Grant 62206189, and the China Postdoctoral Science Foundation under Grant 2023M732427.

## Impact Statement

This paper proposes a novel perspective for viewing U-Nets and a new method for handling skip connections, which can be widely applied to U-Nets. We hope that this research can contribute to the advancement of medical image segmentation. We do not think this work has any harmful impact.

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

# Appendix

## A. The detailed process of updating the memory flow Y.

The first time we update the memory stream, i.e., derive $Y_2$, we use a one-step implicit approach.

**Theorem A.1. *one-step Adams–Moulton method* (*Moulton, 1928*).** *Given the derivative $\dot{Y}_t = F_t$, let $t = t_n$, and choose a step size $\delta$ along the t-axis such that $t_{n+1} = t_n + \delta$. The approximation of $Y_{t_{n+1}}$ at the next integration point is given by:*

$$Y_{t_{n+1}} = Y_{t_n} + \frac{\delta}{2} \cdot (F_{t_n} + F_{t_{n+1}}). \tag{5}$$

For a U-Net with $L$ stages, let $X_l$ and $Y_l$ be $X_{t_n}$ and $X_{t_n}$ at different time, respectively, $1 \leq l \leq L$, and set $\delta = 1/L$. Taking $X_{t_n} = X_1$ and $Y_{t_n} = Y_1$, based on **Theorem A.1**, we have:

$$Y_2 = Y_1 + \frac{\delta}{2} \cdot (F_1 + F_2). \tag{6}$$

In the process of calculating $Y_2$, the term $F_2 = F(X_2, Y_2)$ is unknown. Therefore, we first need to predict the value of $Y_2$, substitute it into $F$ to compute $F_2$, and then correct the value of $Y_2$. According to the description in Section 3.3, second paragraph, the known values at this stage are $X_2$, $X_1$, $Y_1$, and $F_1$. When predicting $Y_2$, we can only use a one-step explicit method.

**Theorem A.2. *one-step Adams–Bashforth method* (*Bashforth & Adams, 1883*).** *Given the derivative $\dot{Y}_t = F_t$, let $t = t_n$, and choose a step size $\delta$ along the t-axis such that $t_{n+1} = t_n + \delta$. The approximation of $Y_{t_{n+1}}$ at the next integration point is given by:*

$$Y_{t_{n+1}} = Y_{t_n} + \delta \cdot F_{t_n}. \tag{7}$$

According to **Theorem 3.2** and **Theorem A.2**, we can obtain the predicted value of $Y_2$,i.e., $\overline{Y_2}$ and calculate $F_2$.

$$\begin{aligned} \overline{Y_2} &= Y_1 + \delta \cdot F_1 \\ F_2 &= F(X_2, \overline{Y_2}). \end{aligned} \tag{8}$$

At this point, we have obtained all the information necessary to compute $Y_2$. In this process, we obtain $Y_2$ and $F_2$. The next step is to compute $Y_3$ and $F_3$ using the two-step implicit method.

**Theorem A.3. *two-step Adams–Moulton method* (*Moulton, 1928*).** *Given the derivative $\dot{Y}_t = F_t$, let $t = t_n$, and choose a step size $\delta$ along the t-axis such that $t_{n+1} = t_n + \delta$, $t_{n+2} = t_n + 2\delta$. The approximation of $Y_{t_{n+2}}$ is given by:*

$$Y_{t_{n+2}} = Y_{t_{n+1}} + \frac{\delta}{12} \cdot (5F_{t_{n+2}} + 8F_{t_{n+1}} - F_{t_n}). \tag{9}$$

In the U-Net specified before Eq. 6, according to Theorem A.3, we have:

$$Y_3 = Y_2 + \frac{\delta}{12} \cdot (5F_3 + 8F_2 - F_1). \tag{10}$$

Similarly, $F_3$ is unknown, the known values at this stage are $X_3$, $X_2$, $X_1$, $Y_2$, $Y_1$, and $F_2$, $F_1$. Now we can use a two-step explicit method to predict $Y_3$.

**Theorem A.4. *two-step Adams–Bashforth method* (*Bashforth & Adams, 1883*).** *Given the derivative $\dot{Y}_t = F_t$, let $t = t_n$, and choose a step size $\delta$ along the t-axis such that $t_{n+1} = t_n + \delta$, $t_{n+2} = t_n + 2\delta$. The approximation of $Y_{t_{n+2}}$ is given by:*

$$Y_{t_{n+2}} = Y_{t_{n+1}} + \frac{\delta}{2} \cdot (3F_{t_{n+1}} - F_{t_n}). \tag{11}$$

According to **Theorem 3.2** and **Theorem A.4**, we can get $\overline{Y_3}$ and $F_3$.

$$\overline{Y_3} = Y_2 + \frac{\delta}{2} \cdot (3F_2 - F_1)$$
$$F_3 = F(X_3, \overline{Y_3}).$$

(12)

Then we finish the calculation of Eq. 10, next step is to compute $Y_4$ and $F_4$ using the three-step implicit method.

**Theorem A.5.** *three-step Adams–Moulton method (Moulton, 1928). Given the derivative $\dot{Y}_t = F_t$, let $t = t_n$, and choose a step size $\delta$ along the $t$-axis such that $t_{n+1} = t_n + \delta$, $t_{n+2} = t_n + 2\delta$, $t_{n+3} = t_n + 3\delta$. The approximation of $Y_{t_{n+3}}$ is given by:*

$$Y_{t_{n+3}} = Y_{t_{n+2}} + \frac{\delta}{24} \cdot (9F_{t_{n+3}} + 19F_{t_{n+2}} - 5F_{t_{n+1}} + F_{t_n}).$$

(13)

In the U-Net specified before Eq. 6, according to Theorem A.5, we have:

$$Y_4 = Y_3 + \frac{\delta}{24} \cdot (9F_4 + 19F_3 - 5F_2 + F_1).$$

(14)

$F_4$ is unknown, the known values at this stage are $X_1 : X_4, Y_1 : Y_3$ and $F_1 : F_3$. Now we can use a three-step explicit method to predict $Y_4$.

**Theorem A.6.** *three-step Adams–Bashforth method (Bashforth & Adams, 1883). Given the derivative $\dot{Y}_t = F_t$, let $t = t_n$, and choose a step size $\delta$ along the $t$-axis such that $t_{n+1} = t_n + \delta$, $t_{n+2} = t_n + 2\delta$, $t_{n+3} = t_n + 3\delta$. The approximation of $Y_{t_{n+3}}$ is given by:*

$$Y_{t_{n+3}} = Y_{t_{n+2}} + \frac{\delta}{12} \cdot (23F_{t_{n+2}} - 16F_{t_{n+1}} + 5F_{t_n}).$$

(15)

According to **Theorem** 3.2 and **Theorem** A.6, we can get $\overline{Y_4}$ and $F_4$ as follows:

$$\overline{Y_4} = Y_3 + \frac{\delta}{12} \cdot (23F_3 - 16F_2 + 5F_1)$$
$$F_4 = F(X_4, \overline{Y_4}).$$

(16)

Finally, the computation of Eq. 14 is completed. When $4 < t_{n+3} \leq L$, we still use Eq. 13 to calculate $Y_{t_{n+3}}$. For example, $Y_5 = Y_4 + \frac{\delta}{24} \cdot (9F_5 + 19F_4 - 5F_3 + F_2)$. However, at this stage, we have more known information. When predicting $Y_5$, we can apply a four-step explicit method, unlike the prediction of $Y_4$, where only a three-step explicit method can be used.

**Theorem A.7.** *four-step Adams–Bashforth method (Bashforth & Adams, 1883). Given the derivative $\dot{Y}_t = F_t$, let $t = t_n$, and choose a step size $\delta$ along the $t$-axis such that $t_{n+1} = t_n + \delta$, $t_{n+2} = t_n + 2\delta$, $t_{n+3} = t_n + 3\delta$, $t_{n+4} = t_n + 4\delta$. The approximation of $Y_{t_{n+4}}$ is given by:*

$$Y_{t_{n+4}} = Y_{t_{n+3}} + \frac{\delta}{24} \cdot (55F_{t_{n+3}} - 59F_{t_{n+2}} + 37F_{t_{n+1}} - 9F_{t_n}).$$

(17)

According to **Theorem** 3.2 and **Theorem** A.7, we can get $\overline{Y_5}$ and $F_5$ as follows:

$$\overline{Y_5} = Y_4 + \frac{\delta}{24} \cdot (55F_4 - 59F_3 + 37F_2 - 9F_1)$$
$$F_5 = F(X_5, \overline{Y_5}).$$

(18)

In the subsequent computations, we perform calculations based on **Theorem** A.5 and A.7. When the computation reaches the topmost stage, the known information includes $X_1 : X_L, Y_1 : Y_L$ and $F_1 : F_L$. At this point, calculating $Y_{final}$ cannot apply the implicit method because $X_{L+1}$ does not exist for the calculation. Therefore, we switch to a four-step explicit method. Then we have:

$$Y_{final} = Y_L + \frac{\delta}{24} \cdot (55F_L - 59F_{L-1} + 37F_{L-2} - 9F_{L-3}).$$

(19)

Finally, we apply a convolution layer with a kernel size of 1 to map $Y_{final}$ to the prediction map.

*Table 6.* Workflow and Results. Take a network with 6 stages as an example for demonstration. P, C, Cal, F stand for Predictor, Corrector, Calculator, nmODEs block, respectively. $F_i = -Y_i + f(Y_i + g(X_i))$.

| Source | Workflow | Result |
|---|---|---|
| $X_1, Y_1$ | P: $Y_2 = Y_1 + \delta \cdot F_1$ 
 C: $Y_2 = Y_1 + \frac{\delta}{2} \cdot (F_1 + F_2)$ | $Y_2$ |
| $X_{1:2}, Y_{1:2}$ | P: $Y_3 = Y_2 + \frac{\delta}{2} \cdot (3F_2 - F_1)$ 
 C: $Y_3 = Y_2 + \frac{\delta}{12} \cdot (5F_3 + 8F_2 - F_1)$ | $Y_3$ |
| $X_{1:3}, Y_{1:3}$ | P: $Y_4 = Y_3 + \frac{\delta}{12} \cdot (23F_3 - 16F_2 + 5F_1)$ 
 C: $Y_4 = Y_3 + \frac{\delta}{24} \cdot (9F_4 + 19F_3 - 5F_2 + F_1)$ | $Y_4$ |
| $X_{1:4}, Y_{1:4}$ | P: $Y_5 = Y_4 + \frac{\delta}{24} \cdot (55F_4 - 59F_3 + 37F_2 - 9F_1)$ 
 C: $Y_5 = Y_4 + \frac{\delta}{24} \cdot (9F_5 + 19F_4 - 5F_3 + F_2)$ | $Y_5$ |
| $X_{2:5}, Y_{2:5}$ | Cal: $Y_6 = Y_5 + \frac{\delta}{24} \cdot (55F_5 - 59F_4 + 37F_3 - 9F_2)$ | $Y_6$ |

## B. Experimental hyperparameters

*Table 7.* Initial Learning Rate Settings

| | ACDC | KiTS | MSD | ISIC2017 | ISIC2018 |
|---|---|---|---|---|---|
| Backbone | 1e-2 | 3e-4 | 1e-4 | 1e-3 | 1e-3 |
| FuseUNet | 3e-2 | 1e-3 | 2e-4 | 3e-3 | 3e-3 |

## C. Computational Cost Analysis

*Table 8.* Computational Cost Analysis for FuseUNet-Ori Decoder and Skip Connection

| FuseUNet - Ori | Decoder | Skip Connection |
|---|---|---|
| Params | $L \cdot 4N^2 \cdot 1^2 - \sum_{i=L}^{1} 3C_i^2 \cdot k^2$ | $\sum_{i=L}^{1} 2N * C_i \cdot 1^2 - 0$ |
| Flops | $2L \cdot 4N^2 \cdot 1^2 \cdot H_o \cdot W_o - \sum_{i=L}^{1} 6C_i^2 \cdot k^2 \cdot H_i \cdot W_i$ | $2\sum_{i=L}^{1} 2N * C_i \cdot 1^2 \cdot H_o \cdot W_o + 7H_o \cdot W_o \cdot 2N - 0$ |

Taking the L-stage convolutional architecture as an example. In the table, N and the superscript 'o' represent the number of target classes and the original, respectively. The reduction in parameters and computation is tied to the number of channels in each stage. Flops increase mainly due to interpolation in skip connections, especially when the original network has fewer channels.

*Table 9.* Detailed comparison of the computational cost data.

| VRAM (G) / Epoch (s) | nn-UNet | UNETR | UltraLight VM-UNet |
|---|---|---|---|
| Backbone | 7.33/144 | 9.4/115 | 0.87/21 |
| FuseUNet | 5.91/128 | 7.8/105 | 1.27/22 |

# D. Detailed performance

*Table 10.* Detailed performance data for each fold on 3D tasks

| Dataset | Fold | Dice1 | Dice2 | Dice3 | Dice avg |
|---------|------|-------|-------|-------|----------|
| ACDC    | 1    | 90.37 | 90.46 | 94.60 | 91.81 |
| 1-Myo   | 2    | 91.21 | 89.40 | 94.49 | 91.70 |
| 2-RV    | 3    | 89.32 | 90.11 | 94.56 | 90.33 |
| 3-LV    | 4    | 91.58 | 90.28 | 94.32 | 92.06 |
|         | 5    | 88.41 | 90.02 | 94.52 | 90.98 |
| KiTS23  | 1    | 97.28 | 83.94 | 79.51 | 86.91 |
| 1-Kidney| 2    | 95.49 | 80.86 | 79.67 | 85.34 |
| 2-Cyst  | 3    | 97.15 | 82.83 | 77.69 | 85.89 |
| 3-Tumor | 4    | 96.89 | 87.11 | 82.82 | 88.94 |
|         | 5    | 95.55 | 84.58 | 75.48 | 83.87 |
| MSD     | 1    | 77.92 | 60.81 | 76.53 | 71.75 |
| 1-WT    | 2    | 79.72 | 61.12 | 79.42 | 73.42 |
| 2-ET    | 3    | 81.40 | 60.53 | 78.23 | 73.40 |
| 3-TC    | 4    | 77.92 | 59.47 | 79.72 | 72.37 |
|         | 5    | 80.56 | 58.68 | 76.79 | 72.01 |

*Table 11.* Model Performance Across Datasets of Fig. 5.

| Order | KiTS | ACDC | MSD | ISIC2017 | ISIC2018 |
|-------|------|------|-----|----------|----------|
| 1 | 84.8 | 91.75 | 71.22 | 89.25 | 88.63 |
| 2 | 85.2 | 91.84 | 71.49 | 89.65 | 89.06 |
| 3 | 85.7 | 91.85 | 71.56 | 90.15 | 89.35 |
| 4 | 86.7 | 92.05 | 71.75 | 90.69 | 89.78 |

*Table 12.* Statistical Analysis for FuseUNet - Backbone

| FuseUNet - Backbone | Mean of Differences | Standard Deviation of Differences | Standard Error of the Mean | 95% Confidence Interval | t-statistic | Degrees of Freedom | p-value |
|---------------------|---------------------|-----------------------------------|----------------------------|-------------------------|-------------|--------------------|---------|
| ACDC | 0.03 | 3.39 | 0.14 | (-0.24, 0.31) | 0.25 | 603 | 0.80 |
| KiTS | 0.15 | 10.18 | 0.27 | (-0.38, 0.67) | 0.55 | 1470 | 0.58 |

The data in Table 12 indicate that the performance of FuseUNet on the ACDC and KiTS datasets shows no statistically significant difference compared to nn-UNet.

# E. Additional Visualization Results

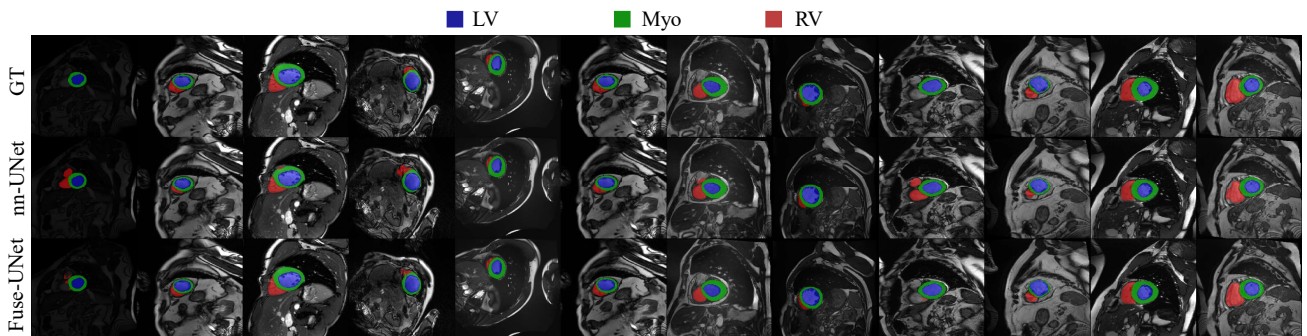

*Figure 6.* Visualization on the ACDC

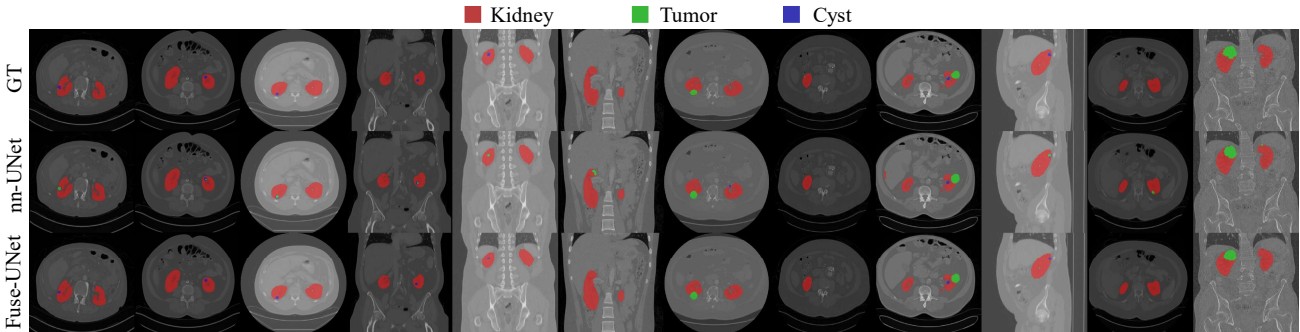

*Figure 7.* Visualization on the KiTS

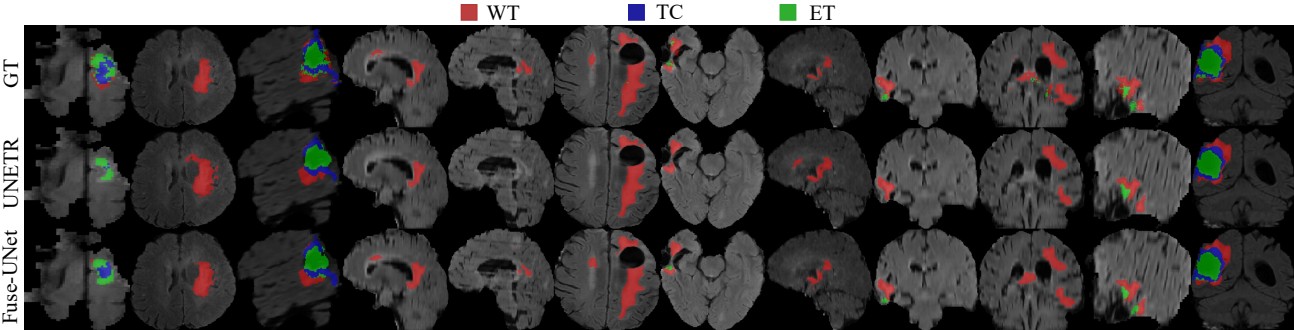

*Figure 8.* Visualization on the MSD

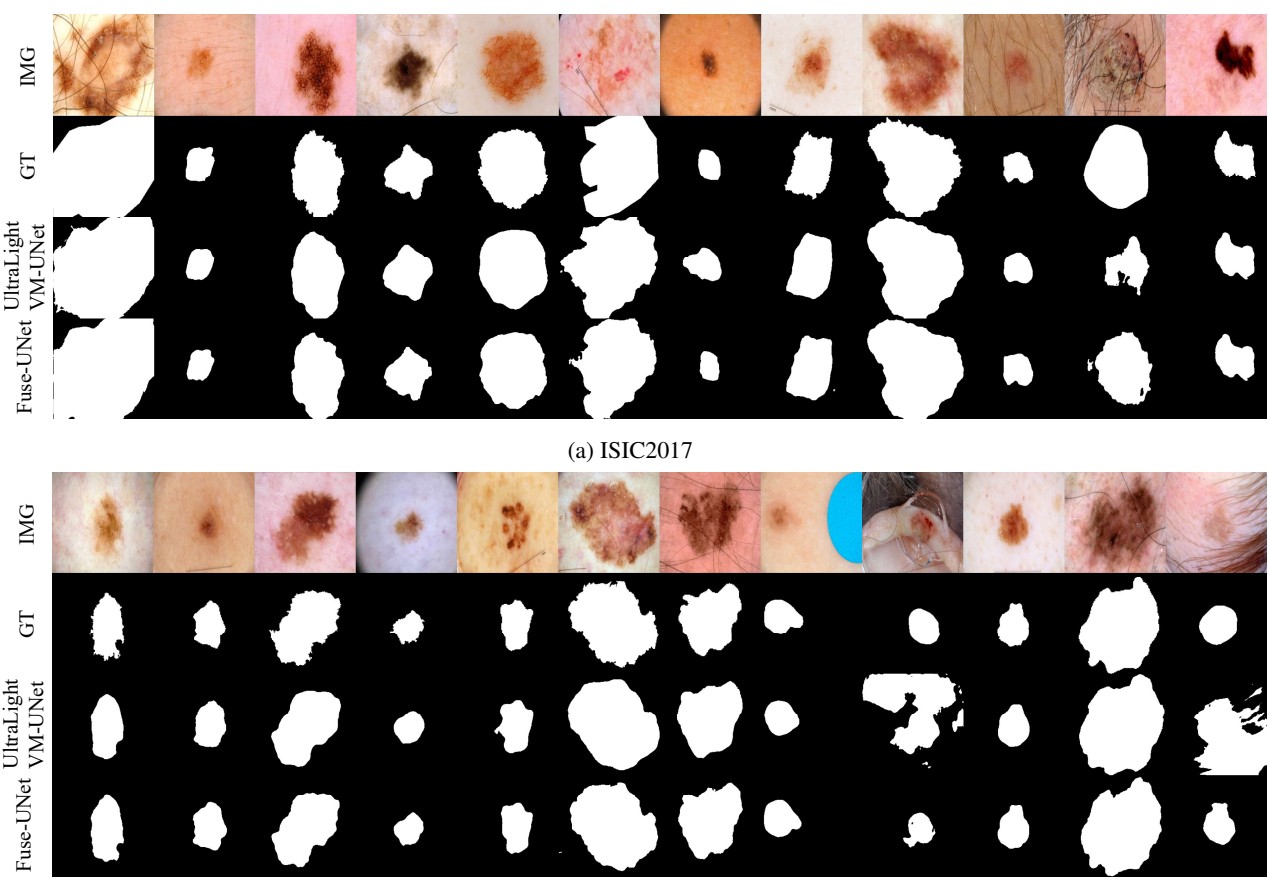

(a) ISIC2017

(b) ISIC2018

*Figure 9.* Visualization on 2D tasks

