# OpenReview forum: "FuseUNet: A Multi-Scale Feature Fusion Method for U-like Networks"
_ICML.cc/2025/Conference — ICML 2025 poster_

### Official Review · Reviewer_n5ho · 2025-03-09

**Overall Recommendation:** 3

**Summary:**

The paper proposes FuseUNet, a multi-scale feature fusion method for U-Net-like networks that enhances skip connection mechanisms by reinterpreting feature fusion as solving an initial value problem (IVP). It employs a linear multistep numerical method with neural memory ODEs (nmODEs) and a predictor-corrector framework, treating skip connections as discrete nodes in an IVP to facilitate effective multi-scale information interaction beyond simple concatenation or addition. Experiments on multiple medical segmentation datasets (ACDC, KiTS2023, MSD brain tumor, ISIC2017, ISIC2018) using CNN, Transformer, and Mamba backbones demonstrate its generalizability, achieving significant reductions in parameters and computational costs while maintaining or surpassing state-of-the-art performance. Ablation studies further analyze the impact of discretization order and memory flow channel numbers, validating the approach’s effectiveness.

**Claims And Evidence:**

1. The paper claims that FuseUNet significantly improves multi-scale feature interaction; however, while experiments show performance comparable or marginally superior to baselines, explicit evidence supporting this claim is lacking. The study primarily reports overall Dice metrics without directly demonstrating that enhanced cross-scale feature interaction is the key factor driving these improvements. Detailed quantitative metrics on multi-scale interaction effectiveness would strengthen the argument.

2. Although the experiments are conducted on diverse architectures, the claim of generalizability across "any U-like network" remains insufficiently validated. The study mainly relies on three specific backbones, making the theoretical generalization claim less convincing. Additional empirical support across a broader range of U-Net variants is needed to substantiate this assertion.

3. The paper highlights a reduction in parameter counts but presents unclear and inconsistent GFLOPs improvements. Notably, for 2D segmentation tasks, GFLOPs slightly increased rather than decreased, contradicting the claimed computational efficiency gains. This discrepancy necessitates a more nuanced explanation or expanded experimental analyses to clarify the impact on computational costs.

**Essential References Not Discussed:**

N/A

**Ethical Review Flag:**

Flag this paper for an ethics review.

**Experimental Designs Or Analyses:**

The chosen benchmarks and backbone networks—CNN-based nn-UNet, Transformer-based UNETR, and Mamba-based UltraLight VM-UNet—are appropriate and widely recognized, providing a solid basis to verify the generality of the proposed method. The authors utilize well-established, publicly available datasets (ACDC, KiTS2023, MSD brain tumor, ISIC2017, ISIC2018), ensuring reproducibility and comparability of results. Additionally, the evaluation protocol employs standard medical image segmentation metrics, including Dice coefficient, sensitivity, specificity, and accuracy, along with five-fold cross-validation, ensuring a robust and reliable assessment.

**Methods And Evaluation Criteria:**

The paper clearly states and conceptually justifies its core methodological innovation: leveraging linear multistep numerical methods, specifically Adams-Bashforth and Adams-Moulton methods, combined with neural memory ordinary differential equations (nmODEs) for multi-scale feature fusion in skip connections. This approach directly addresses the limitations of traditional skip connections in U-Net-based models, making the methodological choice both relevant and theoretically sound for enhancing multi-scale feature interaction.

**Other Comments Or Suggestions:**

N/A

**Other Strengths And Weaknesses:**

1. Marginal Practical Performance Gains: Although the proposed FuseUNet reduces the number of parameters and slightly improves performance in some cases, the practical segmentation improvements (Dice scores) are relatively modest (typically under 1%), especially for well-established benchmarks like ACDC and ISIC datasets. This raises questions about the real-world significance of the proposed method compared to existing models.

2. Limited Justification for Hyperparameter Selection: The authors justify the learning rate adjustments (e.g., increasing learning rates by factors of 2 or 3 due to parameter reduction) but do not provide empirical evidence or systematic hyperparameter tuning results. The lack of thorough justification weakens confidence in whether reported performance gains are optimal or incidental.

3. Insufficient Statistical Validation: The paper does not include statistical significance tests (such as paired t-tests or confidence intervals), leaving uncertainty about whether observed minor differences represent genuine improvements or are within experimental variance. Clarity and Visualization Issues: Certain figures (e.g., Fig. 5 with normalized performance) could be improved for clearer interpretability. Non-normalized, explicit performance metrics would better reflect actual impacts and facilitate easier interpretation by readers.

4. Missing Important Baselines: Although the authors benchmark their method against CNN-based, Transformer-based, and Mamba-based architectures, direct comparisons against closely related skip-connection enhancement approaches, particularly UNet++ and UNet3+, are not sufficiently elaborated upon. These models explicitly address multi-scale fusion, and their detailed comparative evaluation would strengthen the justification of FuseUNet’s claimed advantages.

5. Limited Computational Analysis: The reported GFLOPs increase for the lightweight UltraLight VM-UNet backbone is concerning. Although the authors attribute this to interpolation, no comprehensive computational analysis (e.g., inference time or memory consumption) is provided to clearly demonstrate practical efficiency, limiting a comprehensive assessment of the claimed computational benefits.

6. Weak Demonstration of Generalizability: Despite claims of broad applicability, experiments are limited to standard benchmarks. Results across more diverse medical imaging modalities or different medical scenarios are not presented, limiting the generalizability claims.

**Questions For Authors:**

1.	Statistical Significance of Results: Can the authors provide statistical tests (e.g., paired t-tests or confidence intervals) to verify the statistical significance of the performance improvements reported in Tables 3 and 4? Such tests would clarify whether observed improvements over baseline methods are meaningful or could arise from random fluctuations.
2.	Hyperparameter Justification: Could authors elaborate on how the learning rate and other hyperparameters were selected, ideally presenting ablation studies or grid searches? Clarifying these choices would significantly enhance confidence in the reported performance gains.
3.	Direct Comparison to Alternative Multi-scale Methods: Why were direct comparisons to UNet++ and UNet3+—methods explicitly designed for multi-scale feature fusion—not thoroughly presented or discussed? Can the authors include detailed comparative results to clearly demonstrate FuseUNet’s superiority?
4.	Computational Efficiency and Practical Deployment: Given that FuseUNet shows a minor increase in GFLOPs in lightweight settings, could the authors further quantify inference speed, GPU utilization, and memory consumption explicitly? Detailed metrics would clarify whether FuseUNet is genuinely advantageous for practical deployments.
5.	Generalizability and Broader Applicability: Have the authors tested or considered their method on modalities beyond CT and MRI segmentation tasks, such as ultrasound or pathology images? Providing additional data or experiments would greatly enhance the strength of claims regarding generalizability.
6.	Theoretical Novelty versus Practical Effectiveness: The paper strongly emphasizes theoretical novelty by relating U-Net architectures to numerical ODE methods. Considering the modest practical improvements observed, could the authors clarify whether their primary intent was theoretical innovation (interpretable architecture design) or practical performance improvements?

**Relation To Broader Scientific Literature:**

The paper clearly and accurately positions its contributions within the broader scientific literature, addressing recognized limitations in U-Net-based segmentation architectures (Ronneberger et al., 2015) by enhancing skip connections for improved cross-scale interaction. While existing variants like UNet++ (Zhou et al., 2020) and UNet3+ (Huang et al., 2020) incorporate dense connections and full-scale feature interactions, they primarily rely on simple concatenation or summation, which the authors argue corresponds mathematically to lower-order explicit Euler methods, limiting their information integration capabilities. To overcome this, the authors draw from classical numerical methods—specifically, linear multistep methods such as Adams-Bashforth and Adams-Moulton—and recent advances in neural ordinary differential equations (NODEs, Chen et al., 2018; Yi, 2023). By framing the U-Net decoding process as solving an initial value problem (IVP), the proposed FuseUNet bridges deep learning architecture design with established numerical analysis, enabling higher-order, implicit multi-scale interactions for improved information fusion. This theoretical grounding uniquely situates the work at the intersection of deep learning and classical numerical computation.

**Theoretical Claims:**

The authors conceptualize skip connections as discrete nodes of an initial value problem (IVP), where multi-scale features represent discrete solutions at different timesteps. This analogy is theoretically sound and aligns with established frameworks in neural ordinary differential equations (NODEs) and linear multistep methods. The theoretical background (Section 3.1) on Adams-Bashforth and Adams-Moulton methods, as well as predictor-corrector techniques, is accurate and consistent with classical numerical analysis literature. Additionally, the formulation of neural memory ODEs (nmODEs) in equations (3) and (4) (Section 3.3) aligns with prior theoretical works, correctly capturing the concept of treating neural network decoding steps as discrete ODE solutions. The derivations are presented clearly, with no significant mathematical errors or inconsistencies, demonstrating a careful and rigorous application of established mathematical methods.

---

> ### Author Rebuttal · Authors · 2025-03-30
>
> We sincerely appreciate your recognition of our work and your detailed and constructive comments. Below is a summary of our responses.
>
> ---
>
> ### **Claims and Evidence**
>
> **1. Multi-scale Feature Interaction**
>
> In the first part of the ablation experiment, we compare the performance of different fused scales, highlighting the effectiveness of multi-scale feature fusion. Specific quantitative metrics have yet to be clearly identified in existing papers. We will explore them in future research.
>
> **2. Generalizability**
>
> We acknowledge that evaluating only three architectures is insufficient to justify claims of universal generalizability. We will revise the paper to say "a variety of U-like networks" and clarify the current scope of validation.
>
> **3. GFLOPs**
>
> Here is a theoretical analysis to explain the computational cost trade-offs. Taking the L-stage convolutional architecture as an example. The reduction in parameters and computation is tied to the number of channels in each stage. Flops increase mainly due to interpolation in skip connections, especially when the original network has fewer channels.
>
> |FuseUNet-Ori| decoder|skip connection|
> |-|-|-|
> |Params|$\sum_{i=L}^{1}4N^2\cdot1^2-\sum_{i=L}^{1}3C_i^2\cdot k^2$|$\sum_{i=L}^{1}2N*C_i\cdot1^2-0$|
> |Flops|$2L\cdot 4N^2\cdot1^2\cdot H_o\cdot W_o-\sum_{i=L}^{1}6C_i^2\cdot k^2\cdot H_i\cdot W_i$|$2\sum_{i=L}^{1}2N*C_i\cdot1^2\cdot H_o\cdot W_o+7H_o\cdot W_o \cdot 2N-0$|
>
> N and the superscript 'o' represent the number of target classes and the original, respectively.
>
> ---
>
> ### **Other Strengths and Weaknesses**
>
> **1. Real-world Significance**
>
> Despite modest performance gains on some benchmarks, FuseUNet offers significant computational savings (up to 50% for standard backbones and 30% for lightweight ones), making it advantageous for compute-constrained applications. Besides, FuseUNet is a general framework, not limited to a fixed model, and can be applied to both existing and future models. We also found that some baselines underperformed in our reimplementation, suggesting FuseUNet's benefits may be understated.
>
> We view theoretical innovation and practical performance as complementary. Our method introduces a new perspective on feature fusion using numerical ODEs. In future work, we plan to extend ODE theory to the encoder for further improvements.
>
> **2. Hyperparameter Justification**
>
> We adjusted only the learning rate, keeping other hyperparameters as default. We did not conduct a dedicated ablation study but validated it briefly in the early stage. Hyperparameter tuning is not the focus of this study, and most related works only report them without justification, https://arxiv.org/pdf/2404.09556 simply states "decreasing the learning rate until convergence" without specifying a value.
>
> **3. Statistical Significance**
>
> We added statistical validation as suggested. For Table 4, as metrics were computed from a global confusion matrix over the full test set, per-image scores were unavailable, preventing additional statistical validation. The data shows that FuseUNet performs similarly to the Backbone, which aligns with our claim.
> |FuseUNet-Backbone|Mean of Differences|Standard Deviation of Differences|Standard Error of the Mean|95% Confidence Interval|t-statistic|Degrees of Freedom|p-value|
> |-|-|-|-|-|-|-|-|
> |ACDC|0.03|3.39|0.14|(-0.24,0.31)|0.25|603|0.80|
> |KiTS|0.15|10.18|0.27|(-0.38,0.67)|0.55|1470|0.58|
>
> Fig. 5's performance metrics are shown below.
> |oder|KiTS|ACDC|MSD|ISIC2017|ISIC2018|
> |-|-|-|-|-|-|
> |1|84.8|91.75|71.22|89.25|88.63|
> |2|85.2|91.84|71.49|89.65|89.06|
> |3|85.7|91.85|71.56|90.15|89.35|
> |4|86.7|92.05|71.75|90.69|89.78|
>
> **4. Missing Baselines**
>
> We added suggested baseline where possible, though full evaluation across all datasets may not be feasible within the rebuttal period.
>
> **5. Computational Analysis (Inference & Memory)**
>
> We provided some theoretical analysis in our previous response, and here we offer a detailed comparison of the computational cost data.
>
> |VRAM (G)/epoch (s)|nn-UNet|UNETR|UltraLight VM-UNet|
> |-|-|-|-|
> |Backbone|7.33/144|9.4/115|0.87/21|
> |FuseUNet|5.91/128|7.8/105|1.27/22|
>
> **6. Limited Modality Diversity**
>
> Our work currently demonstrates the generalizability of the proposed method to some extent through segmentation tasks on three data types. Due to time and compute constraints, we didn’t include more modalities but plan to do in future work to further validate our model’s performance across diverse scenarios.
>
> ---
>
> ### **Questions for Authors**
>
> **1–5.** These points are addressed in Other Strengths and Weaknesses - 3, 2, 4, 5, and 6, respectively.
>
> **6. Theoretical vs. Practical Focus**
>
> Our primary contribution is theoretical. By linking U-like networks with numerical ODE methods, we propose a mathematically grounded framework for multi-scale fusion. We hope this new perspective will aid both model design and interpretability, serving as a foundation for future architectural research.

---

> > ### Comment · Reviewer_n5ho · 2025-04-02
> >
> > Thanks to the authors for the detailed response in the rebuttal, which addressed most of my concerns. However, the overall presentation, including formatting and figure aesthetics, falls short of the standards typically expected at ICML. While the strong experimental results support the core claims of the paper, the subpar presentation leaves room for concern. I am currently leaning towards acceptance, but I acknowledge that a rejection could also be justified.

---

> > > ### Author Response · Authors · 2025-04-03
> > >
> > > We sincerely thank you for the thoughtful evaluation and for acknowledging the strength of our experimental results and core contributions. We also appreciate your honest feedback regarding the overall presentation quality, including formatting and figure aesthetics. We fully understand the importance of clear and polished presentation for a high-standard venue like ICML.
> > >
> > > In response to your comments, we will carefully refine the paragraph spacing, line breaks, and layout structure throughout the paper to eliminate excessive blank areas, inconsistent indentation, and misaligned elements.
> > >
> > > For tables and figures, we have consulted numerous accepted ICML papers in recent years and revised our presentation style to better align with community standards. Specifically:
> > >
> > > - **Tables**: We optimized the use of borders and adjusted text alignment; added more informative content to sparsely populated tables to reduce excessive surrounding whitespace; and shaded the backbone and FuseUNet rows to better distinguish them from other entries.
> > > - **Figures**: We adjusted the spacing in segmentation visualizations to prevent boundary confusion; and fixed a rendering issue where thin white lines appeared during image scaling. These updates are aimed at improving both clarity and aesthetic quality.
> > >
> > > In addition, during this revision, we noticed that one of the equations in our previous rebuttal contained a minor typographical error. Specifically, the parameter difference between FuseUNet and convolutional backbone in the decoder was incorrectly denoted as **$\sum_{i=L}^{1}4N^2\cdot1^2-\sum_{i=L}^{1}3C_i^2\cdot k^2$**, whereas it should have been **$L \cdot 4N^2\cdot1^2-\sum_{i=L}^{1}3C_i^2\cdot k^2$**.
> > >
> > > We appreciate your careful review, which motivated us to re-examine both presentation and content more thoroughly. We will incorporate all these enhancements in a later version of the paper. Thank you again for your constructive suggestions, which are invaluable in helping us improve the presentation of our work.

---

### Official Review · Reviewer_YxK1 · 2025-03-10

**Overall Recommendation:** 4

**Summary:**

This paper introduces a new multi-scale feature fusion method for skip connections and for the U-Net framework called FuseUNet, which aims to address the problems of lack the capability for multi-scale information interaction. Specifically, it defines the differential relationship between the skip connections and the corresponding stages. Furthermore, FuseUNet introduces nmODEs for optimization of U-Nets networks, which divides neuron into two parts: learning neuron and memory neuron. Most importantly, the approach proposed by authors is  applicable to any U-like network. Comprehensive experiments are conducted on three datasets to demonstrate its effectiveness.

**Claims And Evidence:**

The claims in the submission are supported by clear evidence.

**Essential References Not Discussed:**

No.

**Experimental Designs Or Analyses:**

The experimental results demonstrate performance on both 2D and 3D datasets, but the comparison methods are limited. There is no comparison with other similar methods based on nmODE, and the ablation study is somewhat insufficient.

**Methods And Evaluation Criteria:**

Yes, the proposed methods make sense for the problem.

**Other Comments Or Suggestions:**

The experimental section needs improvement by including more detailed ablation studies on the Predictor-Corrector, Calculator, and nmODEs block, as well as comparisons with similar methods based on nmODE or Predictor-Corrector.

**Other Strengths And Weaknesses:**

###Strengths###
S1: For the related work Linear Multistep Method, Predictor-Corrector Method and nmODEs are introduced relatively clearly, this paper introduces the above methods to optimize U-Net in a reasonable way.
S2: The experimental results are clear and provide abundant comparative results.
S3: The paper presents a systematic improvement to the U-Nets architecture, enabling network training with fewer parameters and achieving faster speed.

###Weaknesses###
W1: More detailed ablation experiments are needed for the three proposed modules in this paper: Predictor-Corrector, Calculator, and the nmODEs block.
W2: The experiments did not include a comparison with nmODE-Unet[*]. Both approaches improve U-Net using nmODE, and both papers focus on medical image segmentation.
W3：The modules: Predictor-Corrector, Calculator, and the nmODEs, need to be described in more detail in the paper, including how they are implemented. A comprehensive workflow description is not clear, needing more clarifications.

[*]Wang S, Chen Y, Yi Z. nmODE-Unet: A novel network for semantic segmentation of medical images[J]. Applied Sciences, 2024, 14(1): 411

**Questions For Authors:**

The reviewer has no additional questions.

**Relation To Broader Scientific Literature:**

Based on prior related efforts[1, 2], this paper combines them together to reduce network parameters and maintain network performance.

[1] Gragg W B, Stetter H J. Generalized multistep predictor-corrector methods[J]. Journal of the ACM (JACM), 1964, 11(2): 188-209.
[2] Yi Z. nmODE: neural memory ordinary differential equation[J]. Artificial Intelligence Review, 2023, 56(12): 14403-14438.

**Theoretical Claims:**

Yes, the theoretical claims in this manuscript are all correct.

---

> ### Author Rebuttal · Authors · 2025-03-30
>
> We sincerely thank you for the constructive and encouraging comments. We appreciate your recognition of our use of Linear Multistep Methods, the clarity of the experiments, and the improvements to U-like networks. Your feedback has helped us refine the paper. Below we provide detailed responses to your comments.
>
> ### **Other Comments or Suggestions**
>
> **W1. More detailed ablation experiments are needed for the Predictor-Corrector, Calculator, and nmODEs block**
>
> The proposed Predictor-Corrector and Calculator modules are implementations of different orders of linear multistep methods. Thus, the ablation study on multistep orders inherently serves as an ablation of these modules. Since they are tightly coupled with established numerical methods, modifying their mathematical formulation would be inappropriate. Apart from varying the step order to explore the trade-off between information depth and complexity, additional experimental variations are difficult to justify.
>
> The second part of our ablation focuses on the memory space within the nmODEs block. We acknowledge the value of more fine-grained ablations as you suggested. Due to time and computational constraints, we were only able to supplement further experiments on 2D datasets in this round. We plan to explore more comprehensive ablation studies in future work.
>
> | F    | ODE     | Dice17 | SE17 | SP17  | ACC17 | Dice18 | SE18  | SP18  | ACC18 |
> |------|---------|--------|-------|-------|--------|--------|-------|-------|--------|
> | ReLU | nmODEs   | 89.55  | 86.53 | 98.49 | 96.30  | 88.36  | 88.65 | 96.70 | 94.96  |
> | Conv | nmODEs   | 89.14  | 87.04 | 97.73 | 95.43  | 88.52  | 86.16 | 97.66 | 95.18  |
> | PVM  | simple  | 89.18  | 89.59 | 97.46 | 96.01  | 89.25  | 89.50 | 96.96 | 95.35  |
> | PVM  | nmODEs   | 90.69 | 89.59 | 98.20 | 96.62  | 89.77 | 89.10 | 97.41 | 95.62  |
>
> Table: In the ablation study of the nmODEs block, "ReLU" refers to the F function used in https://www.ijcai.org/proceedings/2024/0091.pdf. We also replaced the PVM module with a convolutional block to show the importance of preserving core components. The "simple" variant removes the implicit mapping in the differential equation based on nmODEs. Results suggest that modifying elements of nmODEs block does not improve performance.
>
> ---
>
> **W2. The experiments did not include a comparison with nmODE-Unet**
>
> We did consider comparing with nmODE-Unet. However, the relevant paper does not provide open-source code or sufficient details about the internal design of the nmODE block, which made reproduction infeasible. For related works that are open-source, their methods are essentially equivalent to our first- and second-order approaches, with the main difference lying in the choice of the function \( f \). Based on your suggestion, we have conducted additional experiments on 2D datasets to provide further comparisons and have reported the results accordingly.
>
> ---
>
> **W3. The modules (Predictor-Corrector, Calculator, nmODEs) need clearer descriptions, including implementation and workflow**
>
> While we have designed the Predictor-Corrector, Calculator, and nmODEs modules in detail, the limited space in the main text prevented us from fully presenting their implementation. To address this, we have included step-by-step derivations in the appendix. However, we acknowledge that the textual explanation may lack clarity in linking the mathematical formulation to the module structure and overall workflow. To improve this, we provide a table to illustrate the workflow and highlight the correspondence between equations and components. Additionally, we have released the [source code](https://anonymous.4open.science/r/FuseUNet-3BA3/README.md) to help readers better understand the implementation.
>
> | source | workflow | result |
> |--------|----------|---------|
> | $X_1,Y_1$ |P: $Y_2 = Y_1 + \delta \cdot F_1$||
> ||C: $Y_2= Y_1 + \frac{\delta}{2}\cdot (F_1 + F_2)$| $Y_2$ |
> |$X_{1:2},Y_{1:2}$|P: ${Y_3} = Y_2 + \frac{\delta}{2}\cdot (3F_2 - F_1)$||
> || C: $Y_3 = Y_2 + \frac{\delta}{12}\cdot (5F_3 + 8F_2 - F_1)$|$Y_3$|
> |$X_{1:3},Y_{1:3}$|P: $Y_4 = Y_3 + \frac{\delta}{12}\cdot (23F_3 - 16F_2 + 5F_1)$||
> || C: $Y_4 = Y_3 + \frac{\delta}{24}\cdot (9F_4 + 19F_3 - 5F_2 + F_1)$|$Y_4$|
> |$X_{1:4},Y_{1:4}$|P: $Y_5= Y_4 + \frac{\delta}{24}\cdot (55F_4 - 59F_3 + 37F_2 - 9F_1)$||
> || C: $Y_5 = Y_4 + \frac{\delta}{24}\cdot (9F_5 + 19F_4 - 5F_3 + F_2)$|$Y_5$|
> |$X_{2:5},Y_{2:5}$|Cal: $Y_6= Y_5 + \frac{\delta}{24}\cdot (55F_5 - 59F_4 + 37F_3 - 9F_2)$|$Y_6$|
>
> The process in the table uses a 6-stage U-shaped network as an example.In the table, P, C, Cal, F stand for Predictor, Corrector, Calculator, nmODEs block, respectively. $F_i = -Y_i + f(Y_i+g(X_i))$.
>
> ---
> ### **Other Comments or Suggestions**
>
> We have addressed this suggestion in detail across the responses above.

---

> > ### Comment · Reviewer_YxK1 · 2025-04-02
> >
> > Thanks to the author for the detailed response in rebuttal, which has addressed most of my concerns.  Considering that the substantial experimental results in this paper are sufficient to support their claim, I would like to recommend to Accept this paper.

---

> > > ### Author Response · Authors · 2025-04-02
> > >
> > > We sincerely thank you for the recognition and thoughtful reassessment of our work. We are very pleased that our rebuttal helped clarify the key points and address your concerns. Your constructive feedback throughout the review process has been invaluable in improving the quality of our paper. We deeply appreciate your final recommendation and your support of our work.

---

### Official Review · Reviewer_FpbK · 2025-03-12

**Overall Recommendation:** 5

**Summary:**

A new variant of U-Net involving a new way to fuse features across different scales. It achieves half the compute of nnUNet while matching performance evaluation.

**Claims And Evidence:**

Not supported. See below.

**Essential References Not Discussed:**

STU-Net variant S in terms of adding it in the ablation.

**Ethical Review Concerns:**

None.

**Experimental Designs Or Analyses:**

Not sound. Explained above why.

**Methods And Evaluation Criteria:**

There are multiple evaluation issues for which I have provided a weak reject even though the core technical foundations seem to be strong.

1. nnUNet is missing in the evaluation of MSD and ISIC(s) with no real reason as to why.
2. There's no real reason provided as to why these datasets are chosen. Very recent (and well defined) literature shows why this is important: https://arxiv.org/pdf/2404.09556 & https://arxiv.org/pdf/2411.03670
3. Relaying to 2, reporting just one metric is not enough.
4. If performance in terms of compute is the main innovation (because the scores across model do not seem statistically significant), there should be much more in depth reporting, especially in terms of VRAM usage and run times. These should not be too extravagant to compute in terms of add-on in experimentation times.
5. Why was STU-L not considered, it achieves comparable performance for almost the same param size.
6. Overall, it seems to me that UltraLight VM-UNet can achieve everything the proposed model can achieve in terms of performance evaluation and compute costs.

**Other Comments Or Suggestions:**

1. It is unclear how they achieved the scores for each class. Did they run all the models? If so, it is quite strange that they have matched the exact average values reported here: https://arxiv.org/pdf/2404.09556 where no class -wise scores have been provided. Unless these have been picked from the specific papers (but this is not mentioned anywhere).

2. A personal opinion would be to test only nnUNet and the proposed model on a very strict set of computational ablations to highlight the effectiveness of the method rather than discuss performance scores across various models. I feel those are not even required to highlight the novelty in the work. It also seems that the authors have the necessary compute to do this.

**Other Strengths And Weaknesses:**

The primary (and seems to be the only) weakness is in the evaluation setting.

**Questions For Authors:**

None.

**Relation To Broader Scientific Literature:**

Important. With the designed steps, it can achieve nnUNet's performance for essentially half the compute. However, at the nnUNet level, it can already run on most commodity hardware and the decrease in compute is not enough to deploy on the edge. This is an important point to be noted in terms of real world use cases.

**Theoretical Claims:**

Did not verify.

---

> ### Author Rebuttal · Authors · 2025-03-30
>
> We sincerely thank you for the valuable comments and suggestions, which have significantly helped us improve the clarity and depth of the paper. Below, we provide point-by-point responses to each concern.
>
> ---
>
> ### **Methods and Evaluation Criteria**
>
> **1. Dataset selection and missing benchmarks**
>
> We selected datasets based on those used in the original backbone papers to ensure fair comparison and avoid inconsistencies across literature. The MSD dataset was not mentioned in [nnUNet v24](https://arxiv.org/pdf/2404.09556), but we added experiments comparing data from [nnUNet v18](https://arxiv.org/pdf/1809.10486) following your suggestion. ISIC2017 and ISIC2018 were not included as no original data is available, and nnUNet has not been evaluated on them in any relevant paper we reviewed. Therefore, we did not include them.
>
> |Dataset|Model|Dice1|Dice2|Dice3|Dice avg|
> |-|-|-|-|-|-|
> |MSD|nn-UNet|80.71|62.22|79.07|74.00|
> ||FuseUNet|80.82|61.32|80.15|74.10|
>
> **2. Justification for dataset choices**
>
> Following the principle mentioned above, we balanced data selection with practical compute constraints. For nnUNet, we used two of the three datasets recommended. BTCV, used in UNETR, was reported as unsuitable for comparison in [1], so we ratained MSD. In UltraLight VM-UNet, PH2 was excluded due to its small size (200 images), while ISIC2017 and ISIC2018 were retained.
>
> **3. Reporting only Dice**
>
> Since our primary reference [nnUNet v24] reports only the Dice coefficient, we followed the same format for consistency and fairness. However, we are pleased to provide more detailed metrics of our experiments in later vision.
>
> **4. Compute-related metrics**
>
> We provide VRAM usage and training time comparisons on RTX4090 between backbone models and FuseUNet. However, we would like to emphasize that compute reduction is not our main contribution; the core innovation lies in the skip connection fusion mechanism, detailed in Response 6.
>
> |VRAM (G) / epoch (s)|nn-UNet|UNETR|UltraLight VM-UNet|
> |-|-|-|-|
> |Backbone|7.33/144|9.4/115|0.87/21|
> |FuseUNet|5.91/128|7.8/105|1.27/22|
>
> **5. On STU-Net comparisons**
>
> It is important to note that STU-Net relies on pretrained weights. In our experiments, we deliberately excluded models that require pretraining to minimize external factors beyond architecture itself. This helps us ensure that the improvements come solely from the proposed structural changes. We acknowledge the value of pretrained models and plan to explore them as part of our future work.
>
> **6. Comparison with UltraLight VM-UNet**
>
> While UltraLight VM-UNet is a strong model, FuseUNet differs in both focus and design advantages.
>
> First, the core innovations of them are fundamentally different. UltraLight VM-UNet emphasizes lightweighting with PVM modules, similar to group convolutions, while FuseUNet focuses on multi-scale fusion using a novel view of U-Net stages as discrete ODE nodes and applying techniques like linear multistep and predictor-corrector methods. This enables effective fusion across stages, with lightweighting as a byproduct.
>
> Second, FuseUNet offers better compatibility with backbones and tasks. UltraLight VM-UNet replaces core modules of others with PVM when applying to them, discarding their core innovations. FuseUNet, on the other hand, enhances skip connections in a way that integrates with existing architectures without altering their structure. Additionally, while UltraLight VM-UNet performs well on 2D single-target tasks, its effectiveness in more complex 3D multi-target segmentation tasks is still unproven.
>
> ---
>
> ### **Relation to Broader Scientific Literature**
>
> The goal of FuseUNet is not solely lightweighting. Instead, we propose a general skip connection fusion strategy applicable to both existing and future U-like networks, with practical significance. Besides, major compute reduction would come from lightweighting the encoder and decoder, but we avoided this to isolate the effect of our method. The encoder remains unchanged.  Incorporating ODE-inspired designs into the encoder is part of our future work. We believe that with further development, FuseUNet's efficiency and deployability will improve.
>
> ---
>
> ### **Other Comments or Suggestions**
>
> **1. Class-wise score source**
>
> For nnUNet, class-wise results were obtained via direct email communication with the author. We will clarify this in the revised manuscript.
>
> **2. Focus on nnUNet-only**
>
> Thank you for recognizing the novelty of our work. While detailed ablation on nnUNet would highlight our method's theoretical effect, generalizability is also key. Since skip connections are common across U-like networks, we prioritize verifying our method's effectiveness across architectures. Deeper ablation studies on nnUNet are planned for future work, as validating generalizability consumed computational resources, limiting our ability to conduct these studies in the current work.

---

> > ### Comment · Reviewer_FpbK · 2025-04-02
> >
> > All my queries were addressed. I switch to a full accept!

---

> > > ### Author Response · Authors · 2025-04-02
> > >
> > > We sincerely thank you for the updated evaluation and for your full acceptance of our work. We're truly encouraged to hear that all your concerns were addressed. Your feedback has been instrumental in helping us refine and clarify the paper, and we greatly appreciate the time and thought you invested throughout the review process. Thank you again for your support and recognition.

---

### Decision · Program_Chairs · 2025-05-01

**Decision:**

Accept (poster)

**Comment:**

This paper proposes a novel approach to multi-scale feature fusion in U-Net-like networks, grounded in linear multistep methods and neural memory ODEs. The theoretical formulation is elegant and well-founded, offering a fresh interpretation of skip connections through the lens of numerical analysis. Reviewers appreciated the conceptual innovation and broad applicability.

The main concerns brought up by the reviewers are missing comparisons with closely related baselines (e.g., nmODE-UNet, UNet++), lack of runtime/memory analysis, and limited statistical validation of results. Most of these criticisms were addressed by the authors and seemingly accepted by the reviewers. Therefore, the consensus is that the paper presents a meaningful and technically sound contribution that fits well within ICML.